# Lower Late Development Rate of Acute Respiratory Distress Syndrome in Patients with Lower Mechanical Power or Driving Pressure

**DOI:** 10.3390/diagnostics14171969

**Published:** 2024-09-06

**Authors:** Ya-Chi Lee, Pi-Hua Liu, Shih-Wei Lin, Chung-Chieh Yu, Chien-Ming Chu, Huang-Pin Wu

**Affiliations:** 1Department of Respiratory Therapy, Chang Gung Memorial Hospital, Keelung 20401, Taiwan; yachi621@cgmh.org.tw; 2Graduate Institute of Clinical Medical Sciences, Chang Gung University, Taoyuan 33302, Taiwan; phliu@mail.cgu.edu.tw; 3Division of Endocrinology and Metabolism, Department of Internal Medicine, Chang Gung Memorial Hospital, Taoyuan 33302, Taiwan; 4Department of Thoracic Medicine, Chang Gung Memorial Hospital, Linkou 33305, Taiwan; ec108146@cgmh.org.tw; 5Division of Pulmonary, Critical Care and Sleep Medicine, Chang Gung Memorial Hospital, Keelung 20401, Taiwan; ycc@cgmh.org.tw (C.-C.Y.); rocephen2000@yahoo.com.tw (C.-M.C.); 6Department of Medical Science, College of Medicine, Chang Gung University, Taoyuan 33302, Taiwan

**Keywords:** mechanical power, driving pressure, pneumonia, acute respiratory distress syndrome

## Abstract

For patients on ventilation without acute respiratory distress syndrome (ARDS), there are, as yet, limited data on ventilation strategies. We hypothesized that driving pressure (DP) and mechanical power (MP) may play key roles for the late development of ARDS in patients without initial ARDS. A post hoc analysis of a database from our previous cohort was performed. The mean DP/MP was computed from the data before ARDS development or until ventilator support was discontinued within 28 days. The association between DP/MP and late development of ARDS within 28 days was determined. One hundred and twelve patients were enrolled, among whom seven developed ARDS. Univariate Cox regression showed that congestive heart failure (CHF) history and higher levels of mean MP and DP were associated with ARDS development. Multivariate models revealed that the mean MP and mean DP were still factors independently associated with ARDS development at hazard ratios of 1.177 and 1.226 after adjusting for the CHF effect. Areas under the receiver operating characteristic curves for mean DP/MP in predicting ARDS development were 0.813 and 0.759, respectively. In conclusion, high mean DP and MP values may be key factors associated with late ARDS development. The mean DP had a better predicted value for the development of ARDS than the mean MP.

## 1. Introduction

A key concern in critical care persists for patients on ventilation support but without the appropriate ventilator settings, as it may exacerbate their lung injury. For patients with acute respiratory distress syndrome (ARDS), specific mechanical ventilation setting recommendations include (1) maintaining a low tidal volume ventilation with 4–8 mL/kg of predicted body weight (PBW), (2) an upper limit goal for plateau pressure (Pplat) of 30 cm H_2_O, (3) no use of recruitment maneuvers, and (4) prolonged prone ventilation for patients with an arterial partial pressure of oxygen (PaO_2_)/fraction of inspired oxygen (FiO_2_) ratio < 150 mm Hg [1,2]. However, the limited data on ventilation strategies for patients with respiratory failure who do not meet the criteria for ARDS prompted this study. The only current guideline for patients with sepsis-induced respiratory failure without ARDS recommends using lower instead of higher tidal volume ventilation [2].

Mechanical power (MP) encompasses several key aspects of relevance. First, it reflects static compliance, which is significant in ventilator-induced lung injury (VILI). Second, it considers the overall impact of positive end-expiratory pressure (PEEP). Third, transpulmonary mechanical power increases with respiratory rate (RR). Therefore, MP is a summary variable encompassing all components potentially contributing to VILI, and it exhibits superior predictive value for different outcomes.

A randomized clinical trial on patients in an intensive care unit (ICU) without ARDS who were anticipated to remain intubated for more than 24 h after randomization revealed that implementing a low tidal volume strategy did not yield a higher count of ventilator-free days compared to an intermediate tidal volume strategy [3]. Moreover, driving pressure (DP) and mechanical power (MP) have emerged as superior predictors of outcomes in patients without ARDS [4]. An international post hoc analysis of a multicenter, prospective, observational, international study revealed that among subjects without ARDS, a higher level of DP on the first day of mechanical ventilation was associated with the subsequent development of ARDS [5]. However, it is still unknown if different mechanical ventilation settings after the period of intubation would increase the risk of the subsequent development of ARDS in patients on ventilation without initial ARDS.

Thus, we sought to identify the association between DP/MP and the late development of ARDS within 28 days after admission to an ICU among patients on ventilation with pneumonia but without initial ARDS.

## 2. Materials and Methods

### 2.1. Procedure

This study is a post hoc analysis of a database from our previous consecutively sampled observational cohort of patients with severe sepsis [6,7]. This study was approved by the Institutional Review Board of Chang Gung Memorial Hospital, and the need for written informed consent was waived (202301344B0C501). Patients admitted to the medical ICU at Chang Gung Memorial Hospital, Keelung, Taiwan, from July 2007 to June 2010, due to severe pneumonia were selected. The exclusion criteria included infections other than pneumonia, the absence of invasive ventilator support, an unknown PaO_2_/FiO_2_ ratio, death on the day of admission, N initial ARDS, PaO_2_/FiO_2_ ratio ≤ 300 mm Hg, or death due to septic shock within 7 days of admission to the ICU. None of the included patients withdrew from this study.

### 2.2. Disease Definitions

Pneumonia was defined as a new abnormal infiltration on chest radiograph with respiratory symptoms or fever. Severe pneumonia was defined as pneumonia complicated by acute respiratory failure requiring intubation and mechanical ventilation with or without septic shock [8]. Sepsis and septic shock were defined according to Sepsis-3 guidelines [9]. Sepsis was defined as a suspected or documented infection with acute increase (≥2) in the Sequential Organ Failure Assessment points. Septic shock was defined as sepsis with blood lactate level > 18 mg/dL and hypotension that was unresponsive to fluid resuscitation, requiring vasopressors to maintain mean arterial pressure ≥ 65 mm Hg during the first 3 days following ICU admission. Stage 2 or 3 acute kidney injury was defined according to Kidney Disease Improving Global Guidelines (KDIGO) [10]. Disease severity was assessed with the Acute Physiology and Chronic Health Evaluation (APACHE) II score [11]. ARDS was defined according to the Berlin definition [12]. ARDS was evaluated via chest radiographs obtained after intubation with ventilator support. The development of ARDS was defined as late development of ARDS within 28 days after admission to the ICU.

### 2.3. Ventilator Settings and Weaning in Our Previous Cohort

In our hospital, patients on ventilation receive pressure-targeted ventilation routinely. Following intubation, all patients were treated with pressure-controlled ventilation with a target tidal volume of approximately 10 mL/kg PBW for patients with non-ARDS. The goal was to maintain an inspiratory Pplat of less than 30 cm H_2_O. The PEEP level and FiO_2_ were adjusted to maintain PaO_2_ greater than 60 mmHg or oxygen saturation by pulse oximetry (SpO_2_) greater than 90%. Ventilator settings were adjusted after 2 h of the first setting.

Ventilator weaning and adjustment were performed at regular intervals (every 8 h) and as necessary according to the general weaning guidelines and clinical practice of our respiratory therapy department [13]. Briefly, respiratory therapists screened patients daily for the following weaning criteria:Mean blood pressure greater than 65 mmHg;Heart rate less than 140 beats per minute;SpO_2_ greater than 92%;PEEP less than 8 cm H_2_O;FiO_2_ less than 35%;Pressure support mode ventilation with pressure less than 10 cm H_2_O.

Patients meeting these criteria entered a 2 h spontaneous breathing trial (SBT). If patients successfully passed the SBT, extubation was performed.

### 2.4. Data Utilized from Our Previous Cohort

ICU admission date was considered as Day 1. The following patient data were recorded within 24 h after admission: age, sex, medical history, and APACHE II score. The history of CHF was based on previous echocardiogram reports. Adverse events were recorded within the first 3 days following admission. Arterial blood gases demonstrating the lowest PaO_2_/FiO_2_ ratio were used within 24 h after intubation with ventilator support. DP was defined as the difference between Pplat and PEEP [14]. DP levels were recorded every 8 h every day. Serial mean data of DP, RR, tidal volume (*V*_T_), and PEEP were recorded daily.

### 2.5. MP Calculation

Following a manual inspiratory hold, the pressure curve equilibrated with the alveolar pressure. Pplat was measured after the flow reached zero. MP for pressure-targeted ventilation was calculated every 8 h every day according to the simplified equation [15], using RR, *V*_T_ size (L), DP, and PEEP:DP (cm H_2_O) = Pplat − PEEP;MP (J/min) = 0.098 × RR × *V*_T_ × (DP + PEEP).

### 2.6. Statistical Analysis

Statistical analysis was performed using the Statistical Package for the Social Sciences version 27.0.1 for Mac (IBM Inc., Armonk, NY, USA). Mean *V*_T_, MP, and DP were computed with arithmetic mean of serial levels of *V*_T_, MP, and DP from Day 1 to the day before ARDS was identified. In patients without late development of ARDS, mean variables were computed with arithmetic mean of serial levels from Day 1 until the day without ventilator support. Differences in the continuous variables between the two groups were analyzed using Student’s *t*-test. The categorical variables between the groups were distinguished using the Pearson chi-squared test or Fisher’s exact test. Univariate and multivariate Cox regression model analyses were performed to study the association between ARDS development and all possible variables. Variables that showed significant differences between groups of ARDS development and no ARDS development were included in the Cox regression model. The binary variables included in the model were coded as present or absent. Since DP is included in the equation for MP, DP and MP were used in the multivariate Cox regression model separately. The cut-off value for predicting the development of ARDS was identified according to the analysis of receiver operating characteristic (ROC) curves. A Kaplan–Meier graph was plotted to analyze the probability of ARDS development after ICU admission. Time to development of ARDS between greater and less than the cut-off value was compared using the log rank test. *p* values less than 0.05 were considered statistically significant.

## 3. Results

We screened 493 patients with sepsis (Figure 1). Finally, 112 patients on ventilation with pneumonia without initial ARDS were enrolled in this study, and 381 were excluded. A total of seven patients developed ARDS within 28 days of ICU admission. Table 1 compares the baseline clinical characteristics of the patients with pneumonia between those with the absence and presence of ARDS development. A history of congestive heart failure (CHF) was more common in the group with ARDS development, who had higher mean MP and DP levels than the ‘no-ARDS development’ group.

According to the univariate Cox regression model, CHF, mean MP, and mean DP were associated with ARDS development within 28 days (Table 2). After the adjustment of the effects of CHF, mean MP (hazard ratio [HR], 1.177; 95% confidence interval [CI], 1.021–1.358) and mean DP (HR, 1.226; 95% CI, 1.016–1.479) were still independently associated with ARDS development in Models 1 and 2 of the multivariate analysis. The areas under the ROC curves (AUROCs) predicting ARDS development for the differences in the DP/MP on Day 1 minus the mean DP/MP were 0.790 (*p* = 0.010) and 0.620 (*p* = 0.287), respectively (Figure 2A). The AUROCs for the mean DP and mean MP were 0.813 (*p* = 0.006) and 0.759 (*p* = 0.022), respectively (Figure 2B).

The Kaplan–Meier curves show the possibility of ARDS development within 28 days after ICU admission for different DP groups (Figure 3). Patients with a difference in the DP on Day 1 minus the mean DP of <3 cm H_2_O demonstrated a significantly higher ARDS development rate than those with a difference in the DP on Day 1 minus the mean DP ≥ 3 cm H_2_O (*p* = 0.029). Patients with a mean DP ≤ 15 cm H_2_O had a significantly lower ARDS development rate than those with a mean DP > 15 cm H_2_O (*p* = 0.004).

## 4. Discussion

The incidence of ARDS development in this study was approximately 6%, which is similar to the 5% in a previous large cohort study [16]. According to the multivariate Cox regression analysis, the mean MP and mean DP were independent factors associated with ARDS development after adjusting for a history of CHF. Patients on ventilation set on a higher mean MP or DP had a higher chance of developing late ARDS compared to those set on a lower MP or DP. Furthermore, the difference in MP on Day 1 minus mean MP did not distinguish the late development of ARDS from the ‘no development’ of ARDS in the ROC curves. This indicates that the MP may have less value than the DP in predicting the development of ARDS. Since the results are based on respiratory system mechanics rather than lung mechanics alone, it is difficult to ascertain whether the MP levels are influenced more by the chest wall or by lung compliance. Consequently, the extent of energy dissipation within the lung tissue remains unknown. This ambiguity may explain why the mean MP did not demonstrate superior discrimination ability compared to the mean DP.

Our study found that patients set with lower DP levels demonstrated lower ARDS development rates. In the Kaplan–Meier curves, patients set with DP levels lower than 3 cm H_2_O compared with initial DP levels on Day 1 had a significantly lower ARDS development rate. Patients set with a mean DP less than or equal to 15 cm H_2_O did not develop ARDS in this study. This suggests that patients with pneumonia without initial ARDS on ventilation should have their DP levels adjusted to less than 15 cm H_2_O or DP levels should be decreased as soon as possible with a target of more than 3 cm H_2_O. In the present literature, no study has reported the results of ARDS development using DP as a guide for ventilator settings to manage patients without initial ARDS. Further studies are required to elucidate the causal relationship between DP and ARDS development.

This study did not ascertain whether Vt/PBW affects ARDS development, although, interestingly, two prior meta-analyses showed that low tidal volume ventilation groups tended to have a lower risk for ARDS development [2,17]. The probable cause may be the lack of low tidal volume ventilation in most of the enrolled patients. The mean Vt/PBW levels were approximately 10 mL/kg in both the ARDS development and no-ARDS-development groups. However, *V*_T_ is proportional to DP, and decreasing the DP level results in a lower *V*_T_ as the pressure target ventilation. Thus, our results still support the somewhat weak recommendation of the 2021 surviving sepsis campaign management guidelines of maintaining a low tidal volume for cases of non-ARDS respiratory failure [2].

MP is considered a unifying theoretical measure of energy transference to the respiratory system, which may cause VILI. Although the mean MP was an independent factor for developing ARDS in patients on ventilation without ARDS with pneumonia in this study, the discriminative ability in predicting late ARDS development was poor when using MP differences on Day 1 minus the mean MP in the ROC analysis. Unlike the role of tidal volume, low tidal volume ventilation has survival benefits for patients with ARDS and the potential benefit of preventing ARDS development in patients without initial ARDS [2]. Especially for patients with ARDS with low respiratory compliance, low tidal ventilation is associated with reduced mortality [18,19]. MP is known to be associated with mortality in patients with ARDS, although the causal relationship remains unclear [4,20]. A recent analysis of three randomized clinical trials shows that the MP and DP, but not tidal volume, were associated with 28-day mortality in patients invasively ventilated without ARDS [21]. However, this study has two major limitations. First, there was an obvious selection bias; from approximately 10,000 eligible patients, less than 2000 were included for analysis. Second, patients may have died from non-pulmonary complications because the causes of death were not collected. Based on current evidence, it remains unclear whether to include MP in ventilatory strategies for patients without ARDS on ventilation.

In a meta-regression model, tidal volume, DP, and MP exhibited similar treatment effects on reducing mortality in patients with ARDS [22]. A pooled database study and two post hoc analyses of observational cohorts in patients with ARDS, DP and MP revealed favorable associations with mortality [4,18,20]. All of the above evidence mentioned thus far supports the recommendation of maintaining the ventilator setting with minimal DP and MP in patients with ARDS. Roca et al. found that the DP level on the first day of mechanical ventilation was associated with the later development of ARDS in subjects without ARDS [5]. However, Roca et al.’s study used ventilatory data collected at a single time point (first day) to compute the parameters of interest, and hence its interpretation cannot be extended to accurately reflect the overall exposure to the parameters. Furthermore, the ARDS group had higher initial DP and MP settings in Roca et al.’s study. In our study, the initial ventilator settings on Day 1 were similar between the two groups. The ARDS development group had higher mean DP and MP before ARDS development compared with the no-ARDS-development group. As we used mean variables to represent the mean whole effects during the period before ARDS, our findings more so suggest that in patients without initial ARDS, a higher DP or MP play a role in the cause, but not in the result, of the late development of ARDS.

A discussion of four further limitations is necessary, which relate to the aforementioned limitations. First, this work was a single-center observational study with a limited number of cases, so the results do not necessarily imply causality. Although only seven patients developed ARDS after ICU admission in this study, the statistical analysis in the Cox regression model was significant. Further studies are necessary to validate our findings. Second, the selection bias from the exclusion of non-survivors due to septic shock should be acknowledged. Third, the ventilation variables may have been impacted by the varying levels of sedation among individual patients. Given that this study was not a randomized controlled trial, it was not possible to fully adjust for the influence of sedation levels between the two groups. Fourth, lung edema due to CHF may have contributed to the observation of a lower PaO_2_/FiO_2_ ratio in severe pneumonia. Thus, our finding of CHF being an independent factor for ARDS development despite being a component of fluid overload in the lung is unclear; nevertheless, the diagnosis of ARDS in this study fits the Berlin definition of ARDS.

## 5. Conclusions

Our findings imply that higher mean DP and MP values may be important factors associated with the late development of ARDS in patients on ventilation with pneumonia and without initial ARDS. The mean DP showed better predictive value for the development of ARDS than the mean MP. Adjusting DP levels to less than 15 cm H_2_O is suggested in patients on ventilation without ARDS to prevent the late development of ARDS.

## Figures and Tables

**Figure 1 diagnostics-14-01969-f001:**
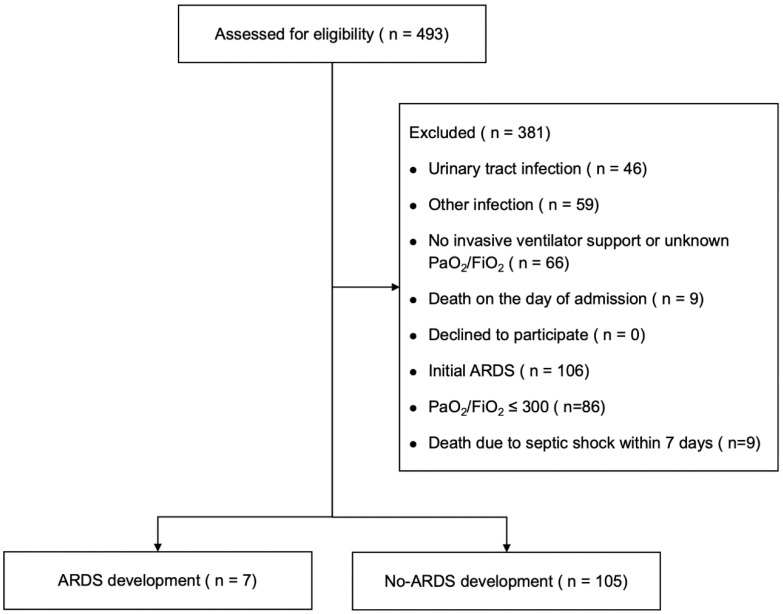
Patient inclusion criteria flowchart. Among 493 screened patients who demonstrated sepsis, 381 were excluded.

**Figure 2 diagnostics-14-01969-f002:**
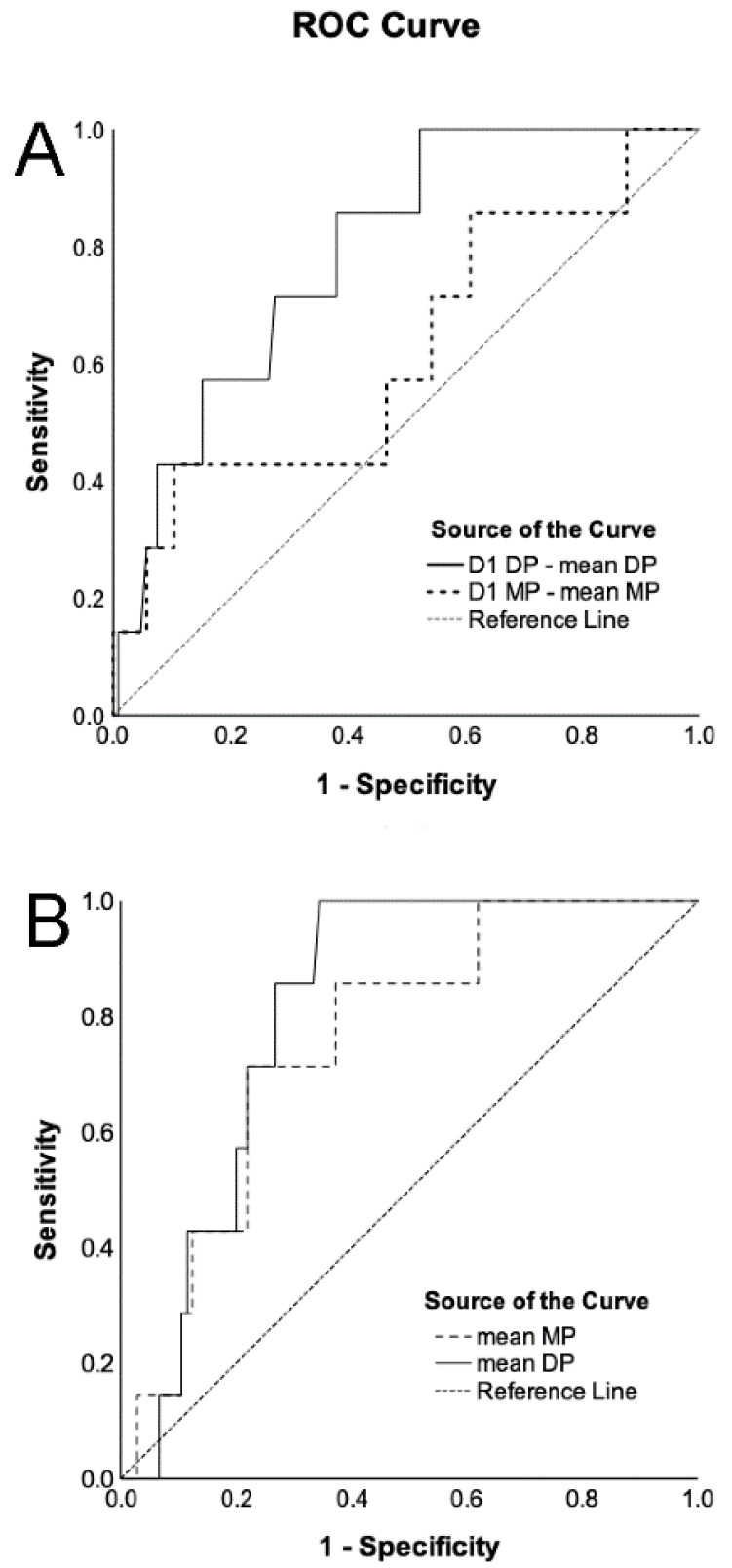
Receiver operating characteristic (ROC) curves of difference of driving pressure (DP) on Day 1 minus mean DP and difference in mechanical power (MP) on Day 1 minus mean MP for the development of acute respiratory distress syndrome (ARDS) in patients without initial ARDS (**A**). The areas under the ROC curves (AUROCs) were calculated. The AUROCs for difference in driving pressure (DP) on Day 1 minus mean DP and difference in mechanical power (MP) on Day 1 minus mean MP were 0.790 (95% confidence interval [CI], 0.649–0.932; *p* = 0.010) and 0.620 (95% CI, 0.387–0.853; *p* = 0.287), respectively. The ROC curves of mean DP and mean MP for the development of ARDS (**B**). The AUROCs for mean DP and mean MP were 0.813 (95% CI, 0.721–0.905; *p* = 0.006) and 0.759 (95% CI, 0.610–0.908; *p* = 0.022), respectively.

**Figure 3 diagnostics-14-01969-f003:**
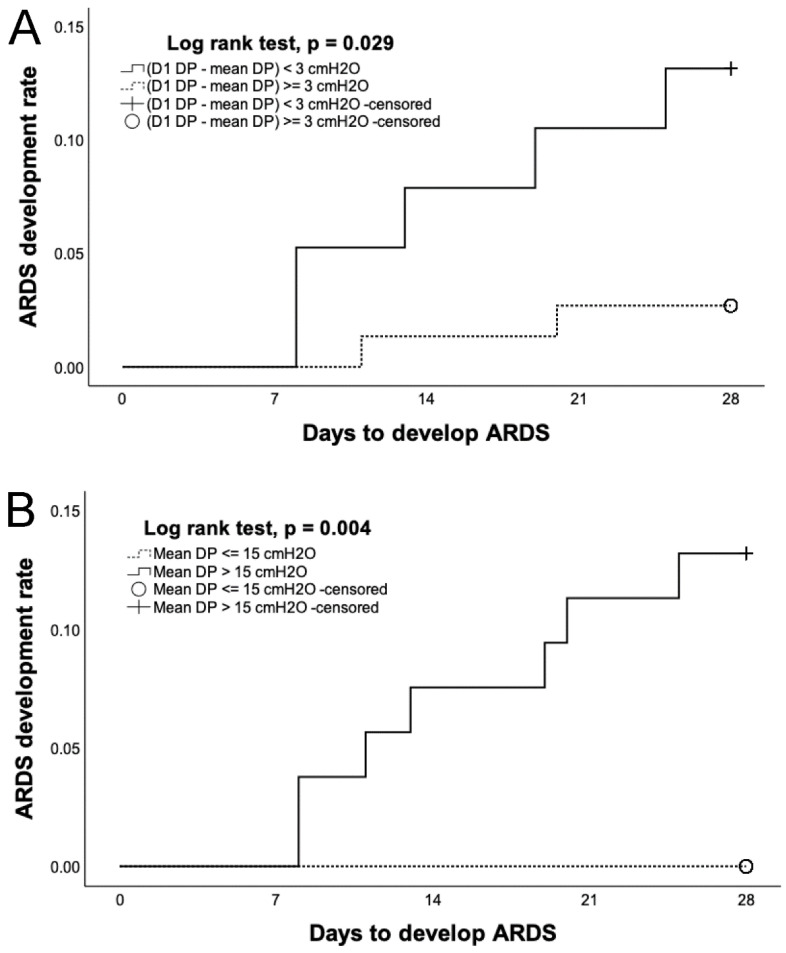
Kaplan–Meier curves of acute respiratory distress syndrome (ARDS) development rate (**A**) within 28 days according to the difference in driving pressure (DP) on Day 1 minus mean DP (≥3, <3 cm H_2_O) and (**B**) within 28 days according to mean DP (≤15, >15 cm H_2_O).

**Table 1 diagnostics-14-01969-t001:** Clinical characteristics of patients with the absence and presence of later ARDS development in patients with pneumonia without initial ARDS on ventilation.

Characteristics	ARDS Development (*n* = 7)	No ARDS Development (*n* = 105)
Age, years	70.3 ± 19.5	77.4 ± 10.8
APACHE II score	29.4 ± 7.1	25.1 ± 6.7
Sex		
Male	5 (71.4)	64 (61.0)
Female	2 (28.6)	41 (39.0)
History		
COPD	1 (14.3)	33 (31.4)
CHF	3 (42.9)	10 (9.5) *
Liver cirrhosis	0 (0.0)	4 (3.8)
Hemodialysis	1 (14.3)	8 (7.6)
Diabetes mellitus	4 (57.1)	36 (34.3)
Murray score on Day 1	1.4 ± 0.3	1.3 ± 0.3
PaO_2_/FiO_2_ ratio (mm Hg)	414.0 ± 96.8	465.5 ± 200.3
Positive end expiratory pressure (cm H_2_O)	7.1 ± 1.9	6.5 ± 1.7
Dynamic lung compliance (mL/cm H_2_O)	29.1 ± 9.6	28.6 ± 8.3
Chest radiography (quadrants infiltrated)	2.0 ± 0.8	1.8 ± 0.8
Adverse events		
Shock	4 (57.1)	30 (28.6)
Stage 2 or 3 acute kidney injury	3 (42.9)	36 (34.3)
GI bleeding	0 (0.0)	9 (8.6)
Thrombocytopenia	2 (28.6)	26 (24.8)
Jaundice	0 (0.0)	0 (0.0)
Ventilator settings on Day 1		
FiO_2_ (%)	60.0 ± 25.5	50.8 ± 22.4
Driving pressure (cm H_2_O)	18.3 ± 3.9	19.5 ± 3.5
Vt/PBW (mL/kg)	9.5 ± 1.6	10.5 ± 2.5
Respiratory rate (/min)	22.1 ± 7.0	20.0 ± 6.3
Mechanical power (J/min)	26.6 ± 10.0	26.6 ± 10.3
Mean Vt/PBW (mL/kg)	10.0 ± 1.0	9.7 ± 1.8
Mean mechanical power (J/min)	24.9 ± 4.4	20.0 ± 5.7 *
Mean driving pressure (cm H_2_O)	18.0 ± 1.6	15.1 ± 3.3 *

Data are shown as mean ± standard deviation and number (percentage). Abbreviations: ARDS = acute respiratory distress syndrome; APACHE = Acute Physiology and Chronic Health Evaluation; COPD = chronic obstructive pulmonary disease; CHF = congestive heart failure; PaO_2_ = arterial partial pressure of oxygen; FiO_2_ = fraction of inspired oxygen; GI = gastrointestinal; Vt = tidal volume; PBW = predicted body weight. * *p* < 0.05 compared with the ARDS development group using *t*-test or chi-squared test.

**Table 2 diagnostics-14-01969-t002:** Cox regression for the analysis of the independent factors for ARDS development.

Variables	Univariate HR (95% CI)	*p* Value	Model 1 *		Model 2 ^†^	
			Multivariate HR (95% CI)	*p* Value	Multivariate HR (95% CI)	*p* Value
CHF	6.435 (1.439–28.789)	0.015	8.064 (1.736–37.456)	0.008	6.523 (1.446–29.432)	0.015
Mean MP (J/min)	1.149 (1.010–1.307)	0.035	1.177 (1.021–1.358)	0.025		
Mean DP (cm H_2_O)	1.208 (1.017–1.435)	0.031			1.226 (1.016–1.479)	0.034

Abbreviations: ARDS = acute respiratory distress syndrome; HR = hazard ratio; CI = confidence interval; CHF = congestive heart failure; MP = mechanical power; DP = driving pressure. * Using history of CHF to adjust mean MP. ^†^ Using history of CHF to adjust mean DP.

## Data Availability

The datasets generated for this study are available on request from the corresponding author.

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
