# Peer review of "Lower Late Development Rate of Acute Respiratory Distress Syndrome in Patients with Lower Mechanical Power or Driving Pressure"

_diagnostics, 2024, doi:10.3390/diagnostics14171969_

Round 1

Reviewer 1 Report

Comments and Suggestions for Authors

Dear Authors,

I found your study's topic “Lower late development rate of acute respiratory distress syndrome in patients with lower mechanical power or driving pressure” to be highly compelling and practically relevant for the management of respiratory failure. The research suggests that high mean driving pressure (DP) and mechanical power (MP) may be key factors associated with late ARDS development. It’s concluded that mechanical ventilation with a DP below 15 may offer a protective effect against the late development of ARDS. In my opinion, the manuscript demonstrates strong overall coherence and quality. However, as you mentioned, the small sample size of ARDS patients and lack of adjustment for confounders are the main limitations of the study. As a result, the findings' validity and reliability should be approached with appropriate caution and require confirmation through larger-scale future investigations with a serious consideration to confounding factors.

Author Response

I found your study's topic “Lower late development rate of acute respiratory distress syndrome in patients with lower mechanical power or driving pressure” to be highly compelling and practically relevant for the management of respiratory failure. The research suggests that high mean driving pressure (DP) and mechanical power (MP) may be key factors associated with late ARDS development. It’s concluded that mechanical ventilation with a DP below 15 may offer a protective effect against the late development of ARDS. In my opinion, the manuscript demonstrates strong overall coherence and quality. However, as you mentioned, the small sample size of ARDS patients and lack of adjustment for confounders are the main limitations of the study. As a result, the findings' validity and reliability should be approached with appropriate caution and require confirmation through larger-scale future investigations with a serious consideration to confounding factors.

Response: Thanks for your suggestions.

Reviewer 2 Report

Comments and Suggestions for Authors

The current article titled “Lower late development rate of acute respiratory distress syndrome in patients with lower mechanical power or driving pressure” Ref: 3102880, deals with an important subject discussing the importance of DP and MP for ARDS patients. Although the hypothesized parameters seems of great usefulness to the patients, the final recommendations should be emphasized, discussed and evidenced supported. The study considered 112 ventilated patients (mild number of studied patients), with pneumonia out of initial 493 patients i.e. 381 patients were excluded (about ¾ of the total number). More revisions should also consider the following items.

- The introduction section should be revised exhibiting the previous reports/publications dealing with the subject of this study. Additionally, the aim of the current study should be emphasized and well explained.

- The conclusion section should be revised supported by the observations of the current study.

- The entire abbreviations should be collected in one list.

Author Response

The current article titled “Lower late development rate of acute respiratory distress syndrome in patients with lower mechanical power or driving pressure” Ref: 3102880, deals with an important subject discussing the importance of DP and MP for ARDS patients. Although the hypothesized parameters seems of great usefulness to the patients, the final recommendations should be emphasized, discussed and evidenced supported. The study considered 112 ventilated patients (mild number of studied patients), with pneumonia out of initial 493 patients i.e. 381 patients were excluded (about ¾ of the total number). More revisions should also consider the following items.

- The introduction section should be revised exhibiting the previous reports/publications dealing with the subject of this study. Additionally, the aim of the current study should be emphasized and well explained.

Response: The introduction section was revised. Previous related reports had been introduced in references 3-5. There was more explanation to mention these studies. To emphasize the aim of this study, there was a separate paragraph to state it.

- The conclusion section should be revised supported by the observations of the current study.

Response: Since the statements in the conclusion section (abstract or main text) were not beyond our results in the current study, the conclusion was not revised.

- The entire abbreviations should be collected in one list.

Response: A abbreviation list was added.

Reviewer 3 Report

Comments and Suggestions for Authors

 This single-center observational study had a limited number of cases, so the results did not necessarily imply causality. Although only  7 patients developed ARDS after ICU admission in this study, the statistical analysis in the Cox regression model was significant, but it was not clear how did the authors correct for the presence of congestive heart failure? Did the authors have BNP,  Lung ultrasound evaluation, or echocardiogram of their patients?   The ventilation variables may have been impacted  by the varying levels of sedation and respiratory efforts among individual patients.  Did the authors measure the inspiratory efforts or transpulmonary driving pressure of their patients?  Lung edema due to CHF may contribute to the observation of lower PaO2/FiO2 ratio in severe pneumonia. Thus, the authors finding of CHF  being an independent factor for ARDS development despite being a component of fluid overload in the lung is unclear; nevertheless, the diagnosis of ARDS in this study fits the  Berlin definition of ARDS. Berlin definition classifies a patient as having ARDS if the patient has no history of CHF or fluid overload, So this study should included only 4 patients with ARDS according to Berlin definition, that was a very limited number of patients. 

Comments on the Quality of English Language

Minor editing of the English language

Author Response

This single-center observational study had a limited number of cases, so the results did not necessarily imply causality. Although only 7 patients developed ARDS after ICU admission in this study, the statistical analysis in the Cox regression model was significant, but it was not clear how did the authors correct for the presence of congestive heart failure? Did the authors have BNP, Lung ultrasound evaluation, or echocardiogram of their patients?

Response: History of CHF was based on previous echocardiogram report. It was added in Method section (Data utilized from our previous cohort). Multivariate Cox regression model analyses were performed to study the association between ARDS development and variables (CHF, mean MP/DP). Since DP is included in the equation of MP, DP and MP were used in multivariate Cox regression model separately, which is shown in Table 2.

The ventilation variables may have been impacted by the varying levels of sedation and respiratory efforts among individual patients. Did the authors measure the inspiratory efforts or transpulmonary driving pressure of their patients?

Response: In this study, the inspiratory efforts or transpulmonary driving pressure were not routinely measured in our patients. The influence of sedation was mentioned in limitations.

Lung edema due to CHF may contribute to the observation of lower PaO2/FiO2 ratio in severe pneumonia. Thus, the authors finding of CHF being an independent factor for ARDS development despite being a component of fluid overload in the lung is unclear; nevertheless, the diagnosis of ARDS in this study fits the Berlin definition of ARDS. Berlin definition classifies a patient as having ARDS if the patient has no history of CHF or fluid overload, So this study should included only 4 patients with ARDS according to Berlin definition, that was a very limited number of patients.

Response: Reviewer might mistake the ARDS diagnosis in the Berlin definition. As the table 3 in JAMA 2012, 307, 2526-2533, ARDS should be excluded by fully explained heart failure or fluid overload. It means that ARDS still can be diagnosed by respiratory failure not fully explained by cardiac failure or fluid overload. In real world, there were many pneumonia patients with CHF history. Why these patients cannot be ARDS if pneumonia progresses? ARDS development group should have 7 patients.